# Spatial and Temporal Variation of PM_10_ from Industrial Point Sources in a Rural Area in Limpopo, South Africa

**DOI:** 10.3390/ijerph16183455

**Published:** 2019-09-17

**Authors:** Cheledi E. Tshehla, Caradee Y. Wright

**Affiliations:** 1Department of Geography, Geoinformatics and Meteorology, University of Pretoria, Pretoria 0002, South Africa; 2Environment and Health Research Unit, South African Medical Research Council, Pretoria 0084, South Africa; Caradee.Wright@mrc.ac.za

**Keywords:** air pollution, point sources, population distribution, particulate matter, TAPM

## Abstract

Air pollution from industrial point sources accounts for a large proportion of air pollution issues affecting many communities around the world. However, emissions from these sources are technically controllable by putting in place abatement technologies with feasible and stringent regulatory conditions in the operation licenses. Pollution from other sources such as soil erosion, forest fires, road dust, and biomass burning, are subject to several unpredictable natural or economic factors. In this study, findings from dispersion modelling and spatial analysis of pollution were presented to evaluate the potential impacts of PM_10_ concentrations from point sources in the Greater Tubatse Municipality of Limpopo, South Africa. The Air Pollution Model (TAPM) was used to model nested horizontal grids down to 10 km for meteorology and 4 km resolution for air pollution was used for simulation of PM_10_. An analysis of annual and seasonal variations of PM_10_ concentrations from point sources was undertaken to demonstrate their impact on the environment and the surrounding communities based on 2016 emissions data. A simple Kriging method was used to generate interpolation surfaces for PM_10_ concentrations from industrial sources with the purpose of identifying their areas of impact. The results suggest that valley wind channeling is responsible for the distribution of pollutants in a complex terrain. The results revealed that PM_10_ concentrations were higher closer to the sources during the day and distributed over a wide area during the night.

## 1. Introduction

Particulate matter (PM) is a complex mixture of solid particles and liquid droplets suspended in the air that vary in size, composition and concentration. PM with aerodynamic diameter of less than 10 µm (PM_10_) can stay in the air for minutes to hours and can travel for up to 50 km, while PM_2.5_ (fine) particles are lighter and can stay in the air for days or weeks and travel longer distances than coarser PM_10_ particles [1]. PM exposure has been a growing public concern and it is associated with severe health impacts [2]. Studies worldwide have linked exposure to PM air pollution with a range of cardiovascular, respiratory diseases, stroke, chronic obstructive pulmonary disease, childhood pneumonia, asthma and diabetes [3,4,5,6,7,8,9]. It should be noted, however, that the evidence for the hazardous nature of combustion-related PM (from both mobile and stationary sources) is more consistent than that for PM from other sources [10]. Ref. [11] estimated that approximately 3% of cardiopulmonary and 5% of lung cancer deaths are attributable to PM exposure globally.

The Greater Tubatse Municipality (GTM) in Limpopo, South Africa, is home to three ferrochrome smelters. The main pollutants emitted from the smelters in the GTM, PM_10_ and PM_10_ chemical components are discussed in [12]. The area has a complex terrain (Figure 1) with terrain height ranging from 850 m to 1850 m above sea level. Mountainous terrain has a high degree of topographical variation and land cover heterogeneity [13]. 

The GTM has an area of approximately 4550 km^2^ in size and a population size of 335,676 [14]. (Table 1) which comprises 115,809 children (0–14) and 17,119 elderlies (65+). These age groups are classified as being prone to the health impacts of PM pollution [15]. The GTM is largely rural with the majority of people being low-income earners and relying on river streams for water and vegetables from informal farming.

Due to intensive industrialization, the area has seen an increase in traffic activities and possibly in waste disposal due to urbanization of this rural area. Due to a lack of ambient air quality data and ground-level meteorological monitoring data in the area, it is vital to investigate how pollution from the industrial point sources impact on the environment. PM is mainly emitted into the atmosphere by anthropogenic sources such as industries, vehicles, combustion sources, road dust and open burnings, and natural sources such as wildfires due to lightning strikes and wind-blown dust from open un-vegetated surfaces.

The emissions alone cannot determine the type and intensity of air pollution of any area. Meteorology and the climate, as well as the topography of the site, all have a major influence on the dispersion and transformation of pollutants [16]. The concentration of pollutants in the lower layers of the atmosphere depends on the atmospheric pressure, the wind and the temperature [17]. The dispersion of pollutants increases with the wind speed and turbulence, and its direction orients the plumes emitted from the stacks. The vertical temperature gradient helps the ascending movement of air pollutants. However, in cases of temperature inversion, pollutants are blocked in the low layers of the atmosphere, which creates episodes of pollution [18].

The topography variation such as the one found in the GTM influences the atmosphere in two ways [19]. The first is in the form of momentum exchange between the atmosphere and the surface that occurs because of flow modification by mountains in the form of mountain lee waves, flow channeling and flow blocking [20]. The second effect involves energy exchange between the terrain and the atmosphere. The thermally induced winds depend on the temperature differences along the mountain plain systems and the strength of the synoptic systems and the cloud cover, with weak synoptic systems and cloud-free atmospheres producing more pronounced winds [21,22]. Mountain winds blow parallel to the longitudinal axis of the valley and are directed up-valley during daytime and down-valley during night-time. The circulation is closed above the mountain ridges by a return current flowing in the reverse direction. The development of thermally driven winds is often complicated by the presence of other wind systems developed on different scales [22,23]. The anabatic flows are limited in time during wintertime as compared to during the summer due to the shorter exposure period to sunlight [24].

Dispersion modelling is a mathematical tool that includes simplified algorithms used to quantify the atmospheric processes that disperse pollutants emitted by a source. Dispersion models can be used to predict concentrations at selected downwind receptor locations depending on the emissions, topographical and meteorological inputs. These models can be used in the development of strategies for the management of pollution impacts on the environment and the assessment of air quality [25]. To effectively manage the PM pollution from industrial sources and to mitigate the impact of this pollutant on human health, it is important to provide information on the spatio-temporal pattern of PM emissions. However, emissions data is not readily available in South Africa.

Ref. [12] identified ferrochrome smelters as one the major contributors of PM_10_ (contributing 23.52% of the total PM_10_ in the GTM). Therefore, an air quality assessment in a highly industrialized rural area of Limpopo, South Africa was performed by means of The Air Pollution Model (TAPM) as a predictive (meteorological) modelling tool. The model output was imported into Geographic Information System to spatially display the seasonal and annual impact of PM concentrations from ferrochrome smelters in the region. The meteorological parameters such as wind, humidity, radiation, mixing height and temperature, were examined to evaluate potential influences on the PM_10_ concentrations.

## 2. Materials and Methods 

### 2.1. Data

The annual PM_10_ emissions data (Table 2) for 2016 were provided by the Department of Environmental Affairs’ National Emissions Inventory System (DEA, NAEIS). The annual data in kilograms per annum (kg/yr) were converted to grams per second for input into the TAPM model.

The population distribution (Table 1) for the GTM was obtained from the Statistics South Africa website [14].

### 2.2. Model Configuration

TAPM (version 4) was run for the period of July 2015 to June 2016. Four grids with nested domains of 25 × 25 horizontal grid points at 30 km (Figure 2a), 20 km (Figure 2b), 10 km (Figure 2c) spacing for the meteorology and 4 km (smaller grid in white) for pollution simulation were used. A total of 49 discrete receptor points were used for calculating pollution concentration fields. The model uses a default database of soil properties, topography, and deep soil parameters were used. The global terrain height data on a longitude/latitude grid at 30-second grid spacing (approximately 1 km) based on public domain data used in the model is available from the US Geological Survey, Earth Resources Observation Systems (EROS) Data Center Distributed Active Archive Center (EDC DAAC). The database for global soil texture types on a longitude/latitude grid at 2-degree grid spacing (approximately 4 km) was obtained from the Food and Agriculture Organization of the United Nation website. The global deep soil parameters data on a longitude/latitude grid at 30-second grid spacing (approximately 1 km) is available from the US Geological Survey, Earth Resources Observation Systems (EROS) Data Center Distributed Active Archive Center (EDC DAAC). 

TAPM has several characteristics: it is three-dimensional, prognostic, Eulerian, incompressible, and non-hydrostatic. It is a primitive equation model in terrain-following coordinates for simulating atmospheric motion and pollutant transport using nested grids [26]. The model comes with databases containing terrain height data, type of soil and vegetation, sea surface temperature and synoptic scale meteorology supplied by the model developer CSIRO Atmospheric Research Australia. Global terrain and land use datasets have a spatial resolution of 1 km. Sea surface temperature data used are monthly averages and have a spatial resolution of 100 km. Meteorological datasets contain six-hourly synoptic scale analyses on a longitude/latitude grid at 0.75- or 1.0-degree grid spacing (approximately 75 km or 100 km). TAPM then ‘zooms-in’ from the 100 km data to model local scales at a finer resolution using a one-way nested approach to improve the efficiency and resolution, predicting local-scale meteorology (typically down to a resolution of 1 km) [27].

TAPM uses the predicted meteorology and turbulence from the meteorological component and consists of four modules. The Eulerian Grid Module (EGM) solves prognostic equations for the mean and variance of concentration. The Lagrangian Particle Module (LPM) can be used to represent near-source dispersion more accurately. Wet and dry deposition effects are also included [27]. In this study the LPM was used for dispersion modelling in the GTM. The model was run in observation mode without data assimilation due to non-availability of meteorological data in the area to assimilate PM exposure.

## 3. Results and Discussion

The simple Kriging method was used to plot the spatial variation of PM_10_ concentrations from the industrial point sources. The method assumes that the mean and variance remain constant and are known in all locations [28]. The meteorological data predicted by the model were extracted at the grid points nearest to the centre of the study area. The annual wind rose (generated by OpenAir statistical software [29]) and spatial variation of PM_10_ concentrations at receptor points were plotted in Figure 3. The annual wind rose shown in Figure 3b indicates that the dominant wind direction is south-westerly, north-easterly and westerly with wind speed reaching 5.7–8.8 m.s^−1^. 

The annual PM_10_ spatial analysis (Figure 3a) indicates that the highest PM_10_ concentration (27.3 µg.m^−3^) was between Smelter 1 and Smelter 2, spreading to the west of Smelter 1 and south-west of Smelter 2. The highest annual concentration was lower than the 40 µg.m^−3^ of the South African PM_10_ National Ambient Air Quality Standards (NAAQS), however, it was higher than the WHO annual mean guideline of 20 µg.m^−3^. Even though the industrial point sources constitute less than a quarter of the total sources in the GTM, they may pose a danger to human health based on the WHO ambient air quality guidelines. The residential areas (indicated in Figure 1b) show areas that are potentially most vulnerable to the impacts of these pollutants from the smelters, given their proximity to the industrial facilities. Prevailing wind speed and direction (Figure 3b) play an important role in determining which areas may be affected by PM_10_ pollution from the point sources, with higher wind speeds dispersing pollutants furthest away from the source and low wind speeds depositing pollutants closest to the sources. The west south-westerly winds are responsible for transporting pollutants to the east north-easterly direction between Smelter 1 and Smelter 2, while the north-easterly and easterly winds are responsible for dispersing pollutants from Smelter 3 to the south-westerly and westerly direction.

Figure 4a and Figure 5a indicate a variation in ambient concentrations during the night-time and day-time, respectively, which is caused by varying meteorological conditions during day time and night time, with night time conditions favouring recirculation of air due to the nocturnal inversion layer, plume impingement on high terrain and persistent wind channeling inside valleys, and the daytime conditions occur mainly when there is vertical mixing when the surface inversion breaks and the winds travel up the slopes and the valley disperse pollutants as a function of time and space, with concentration being higher closer to the sources and lower further away from the sources. The evidence of this is the nonexistence of dark red colours to indicate high concentrations on Figure 4a. However, there is wide spread of higher concentrations during the night that covers large residential areas as compared to high concentration of PM_10_ during the day that are localized closer to the sources. This is a clear indication that there is less mixing during the night compared to the day-time. Therefore, pollution episodes often occur more during the night than during the day. This makes communities more vulnerable to pollution at night than during the day, except those communities residing near the industries. Figure 4a shows re-circulation of PM_10_ during the night, which is distributed west of Smelter 1, north-west of Smelter 2 and west of Smelter 3. The wind profile (Figure 4b and Figure 5b) shows that the easterly winds are the most effective winds that could potentially disperse pollutants to the west of Smelter 1 and Smelter 2.

Figure 6, Figure 7, Figure 8 and Figure 9 show the seasonal distribution of PM_10_ concentrations and wind profiles. The wind profiles show that the dominant wind is west to south-westerly during autumn (Figure 6a,b). These winds are responsible for the distribution of pollutants in a north-easterly direction in the south of the study area. The winds are then channeled in a north-westerly direction toward the north of the study area by following the valley axis. The winter season (Figure 7a,b) is dominated by south-westerly winds. However, the easterly and south-easterly winds are the ones dispersing pollutants to the west and southwest of the facilities. During spring (Figure 8a,b) and summer (Figure 9a,b), the north north-easterly to easterly winds are the dominant winds responsible for the dispersion of pollutants to the west and south-west of the facilities in the area. The south-westerly and north-easterly flows are parallel to the valley axis, particularly near Smelter 1 and Smelter 2. There is little or no wind coming from the southerly sector during all seasons. This could be due to the orientation of the valley axis in a south-westerly and north-easterly direction in the south of the study area and the north-west and south-east in the north of the study area. The lack of winds from the south limits the transport of pollution to the north of the facilities in the area. The seasonal patterns on the PM_10_ concentration distributions show that the north-easterly to easterly winds are responsible for dispersing pollutants to the west and south-west. The seasonal distribution also shows that the highest PM_10_ concentrations varies from 15.2 µg.m^−3^ to 27.3 µg.m^−3^ in the GTM, which constitutes 23.52% of the total PM_10_ sources. These areas likely experience high PM_10_ concentration from industrial point sources and should be considered for passive ambient air quality measurements. This will assist in determining whether there should be further investment in continuous monitoring in these areas. 

The relation between PM_10_, temperature, relative humidity, solar radiation and mixing height were tested using the Openair statistical software in R (Environmental Research Group, King’s College London, Wareloo, England, United Kingdom) [29]. Figure 10 shows the seasonal relationship between PM_10_ and temperature. The autumn, winter and spring PM_10_ and temperature graphs show a D-shape plot with highest temperatures occurring during the day and the lowest concentration occurring in the morning and late afternoon. The winter temperatures reached a minimum of just above 5 °C with a maximum of around 28 °C. The autumn and spring seasons had similar lowest temperature of around 8 °C, however, the spring maximum temperatures were higher than those in autumn. The D-shaped plot indicates that during these seasons, the temperature correlates with PM_10_ concentrations in the GMT. During the summer months, there was a negative correlation between temperature and PM_10_ concentration; the highest PM_10_ concentrations occurred in the morning with minimum concentrations occurring in the evening. These graphs also show that there was a decrease in concentrations when the surface inversion—which normally breaks up early during summer and later during the other seasons—broke. Hence, we see that the PM_10_ concentrations started to decrease at almost the same temperature of about 20 °C in all seasons. The minimum temperatures for summer were around 15 °C, with maximum temperatures reaching up to 36 °C. 

The relation between PM_10_ and relative humidity (Figure 11) indicates that the PM_10_ concentrations increased with increasing humidity, particularly during autumn and summer. During winter and spring, there were high PM_10_ concentrations observed when humidity was low, though this was not as prevalent compared to when humidity was high, as was the case in other seasons. Therefore, humidity conditions positively affected the PM_10_ concentration, whereby moisture particles adhered to PM_10_, accumulating atmospheric PM_10_ concentration.

Figure 12 shows that high PM_10_ concentrations were observed when the mixing height was shallow and decreased with an increasing mixing height. A similar trend was observed for PM_10_ and radiation. This is an indication that the mixing height was dependent on solar radiation. The PM_10_ vertical distribution (Figure 13) was higher in spring and summer and lowest during autumn and winter. This was due to the reduction in the solar radiation that heats the earth’s surface in autumn and winter and more radiating heating reaching the surface during spring and summer due to longer day hours. There is also vertical mixing during spring/summer seasons due to atmospheric instability, while during autumn/winter seasons, there are dominant high-pressure systems prevailing in the southern hemisphere that suppress the mixing height and lead to high concentrations at the surface. The surface inversion during autumn/winter is lower compared to the surface inversion in spring/summer seasons due to higher minimum temperatures in spring/summer compared to autumn/winter minimum temperatures Figure 13 also shows that the maximum PM_10_ concentrations were observed to a depth of up to 500 m above ground level, which is the indication of the boundary layer height in the study area, which therefore implies that the industrial stacks in the area should be raised to a height higher than their current height of less than 100 m to prevent build-up of pollution at the surface. The surface meteorological parameters were not considered in this study due to the non-availability of ground–based monitoring stations. It should be noted that the steep slopes found in the study area can give rise to thermally induced circulations, like mountain–valley breezes, that strongly modify the characteristics of synoptic flow [21,23,30,31].

### PM Seasonl Varaiation

The average PM_10_ concentrations for the different seasons showed maximum values of 15.2 µg.m^−3^ (autumn), 27.3 µg.m^−3^ (winter), 24 µg.m^−3^ (spring), and 25.3 µg.m^−3^ (summer). These results suggest that the potential health risks associated with PM_10_ is low in autumn and high in winter. Although there are no standards for seasonal concentrations, the winter, spring and summer concentrations are high enough to warrant consideration of possible health effects in the GTM, given that these concentrations are from sources that constitute less than a quarter of the total sources in the area.

## 4. Conclusions

PM is a widespread air pollutant that lingers in the atmosphere depending on their size fraction. Pollutants emitted from point sources can be deposited closer to or at a further distance from the sources depending on the wind strength, the atmospheric stability and particle size. This study shows that meteorology played a major role in the distribution of PM_10_ in the study area. The orientation of the valley axis also had an influence on the distribution of pollutants by channelling the wind within the valley axis. The results of the PM_10_ distribution indicate that the concentrations were higher closer to the sources and lower further away from the sources during the day, while higher concentrations were distributed over a wide area during the night. These findings also show that during the day, the communities most likely to be impacted by pollution from the industrial point sources in the study area were those residing closer to the industrial facilities and during the night, the recirculation of pollution will likely impact a number of villages further away from the industries. Depending on the chemical composition, the PM_10_ emitted from the ferrochrome smelters could also carry heavy metals that could potentially impact human health, either directly through inhalation or indirectly through ingestion from water and food harvested from farms in the area depending on the extent of concentrations, given the fact that the simulated annual concentration of 27.3 µg.m^−3^ only constituted 23.52% of the total sources in the area, and was higher than the WHO annual guidelines of 20 µg.m^−3^ [10], though it is lower than the South African annual PM_10_ NAAQS of 40 µg.m^−3^. This finding suggests that South Africa should revise the PM_10_ standards (and implement measures to comply with the set limits) to be comparable to the WHO guidelines in order to provide adequate protection of human health and well-being. Approximately 40% of the population of the GTM falls within the category of vulnerable individuals that are more likely to be impacted by exposure to PM_10_ concentration from the point sources because PM_10_ is a consistent pollutant [10]. It is therefore possible for regulating authorities to put in place stringent measures to reduce pollution from point sources (which are easily regulated compared to other sources of PM pollution). The monitoring of PM_10_ and PM_2.5_ needs to be improved in GTM and South Africa as a whole to assess population exposure and to assist authorities in establishing plans for improving air quality. The results of this study and a study by [12] have been shared with the Limpopo Provincial authorities and the Sekhukhune district municipality authorities. As a result, we have seen both authorities commissioning the ambient air quality monitoring stations in the area. There is also a recommendation to establish an environmental forum that will involve all stakeholders in the area to better manage air pollution sources.

The study used the TAPM model to identify the areas that are most likely to be impacted by emissions from point sources in a complex terrain. Therefore, future studies should consider using meteorological models to simulate wind profiles in a complex terrain and compare the output with the analysis of ground measured meteorological parameters to verify the model simulations. This will help in analyse the statistically relevant dispersion conditions and search for critical situations with regards to the type of model to be used and the pollutant sources concerned. It would therefore be proper for South African regulations regarding dispersion modelling to be amended to include a requirement to perform wind field modelling using mass consistent models in complex terrain prior to performing a dispersion modelling, and to recommend the use of non-hydrostatic models (such as TAPM) on complex terrain.

### Recommendations

TAPM should be considered as a regulating dispersion modelling tool in South Africa because it is now included in the EPA’s list of recommended air dispersion models. There is a need for the DEA to partner with the South African Weather Service to identify areas where there is a need to monitor surface meteorology, particularly in complex terrain. This data will be useful as input when modelling a pollution in complex terrain. Furthermore, both wet and dry deposition of PM_10_ using the TAPM model need to be undertaken to check what the model predicts and compare with the measurements from the new commissioned ambient air quality monitoring stations. Future modelling studies in the area should also involve emissions data from sources identified by [12] which are omitted in this study due to non-availability of emissions data in the area.

## Figures and Tables

**Figure 1 ijerph-16-03455-f001:**
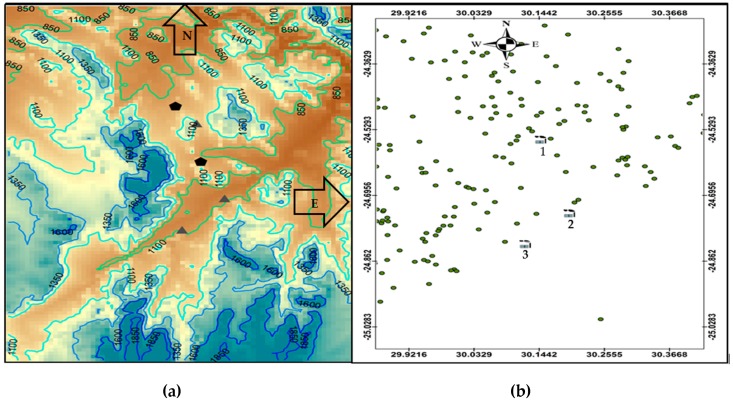
(**a**) Terrain map of the study area showing smelters as black triangles and mine tailings as black pentagons. (**b**) Study area showing location of smelters (1–3) and residential areas in green dots.

**Figure 2 ijerph-16-03455-f002:**
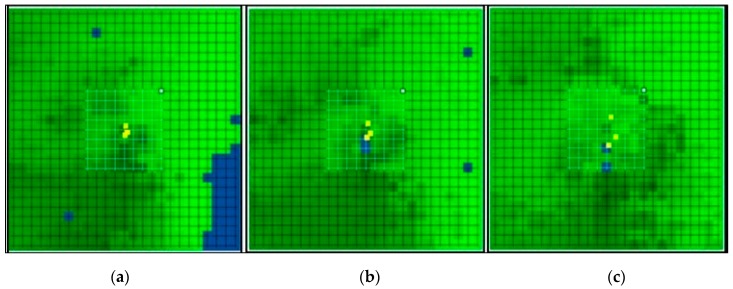
The horizontal grid domain used by TAPM for meteorology and the smaller grid for PM_10_ predictions (25 × 25). These domains were simulated at horizontal grid resolutions of (**a**) 30 km, (**b**) 20 km, (**c**) 10 km and 4 km for PM_10_ emissions.

**Figure 3 ijerph-16-03455-f003:**
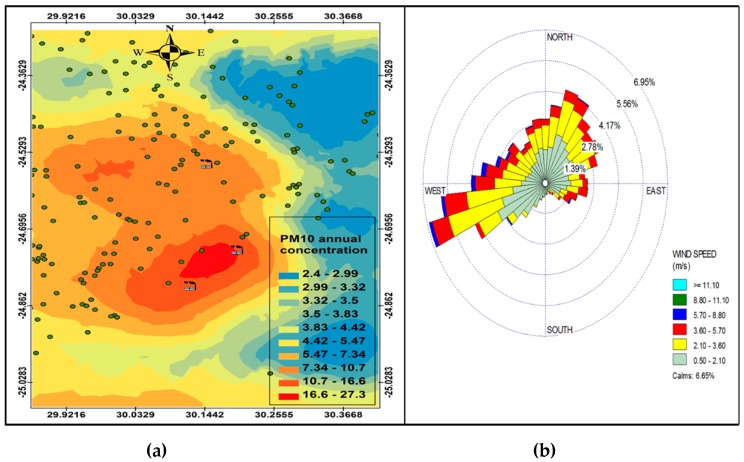
Annual variation in PM_10_ (µg.cm^−3^) ambient concentration (**a**) and annual wind rose (**b**).

**Figure 4 ijerph-16-03455-f004:**
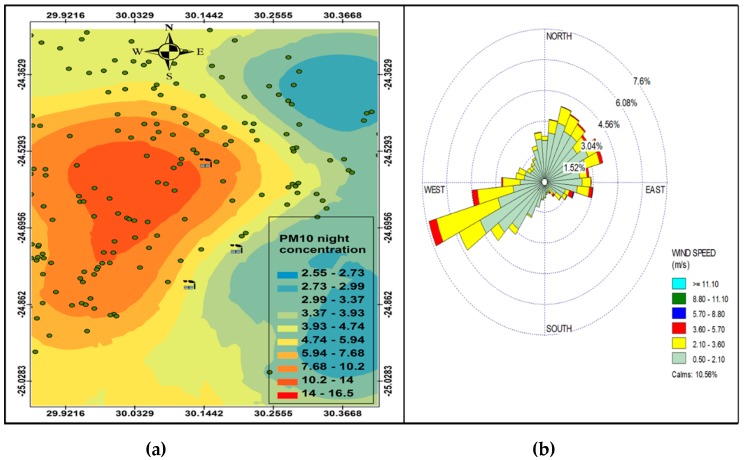
Annual variation in PM_10_ (µg.cm^−3^) concentration at night (**a**), wind rose during the night (**b**).

**Figure 5 ijerph-16-03455-f005:**
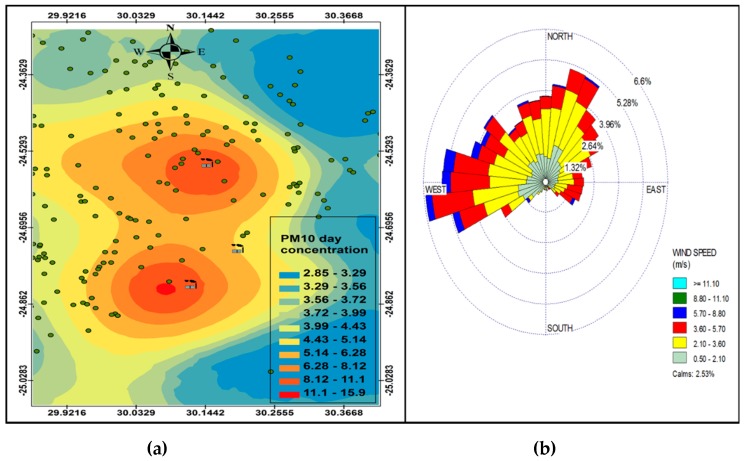
Annual variation in PM_10_ (µg.cm^−3^) concentration during day-time (**a**), wind rose during the day (**b**).

**Figure 6 ijerph-16-03455-f006:**
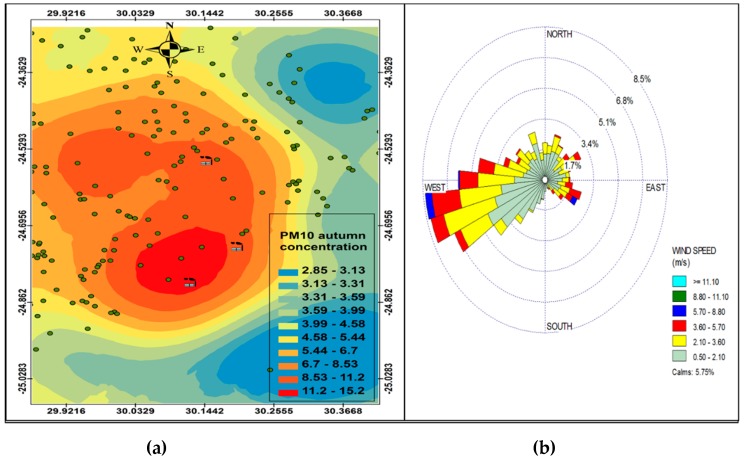
Autumn variation in PM_10_ (µg.cm^−3^) ambient concentration (**a**) and wind roses (**b**).

**Figure 7 ijerph-16-03455-f007:**
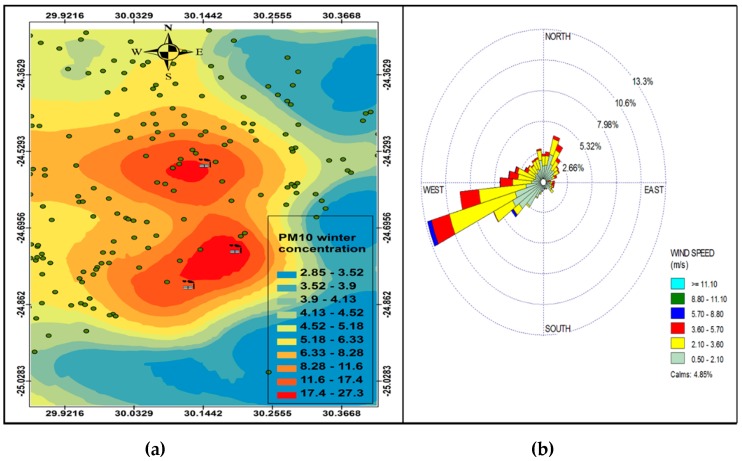
Winter variation in PM_10_ (µg.cm^−3^) ambient concentration (**a**) and wind roses (**b**).

**Figure 8 ijerph-16-03455-f008:**
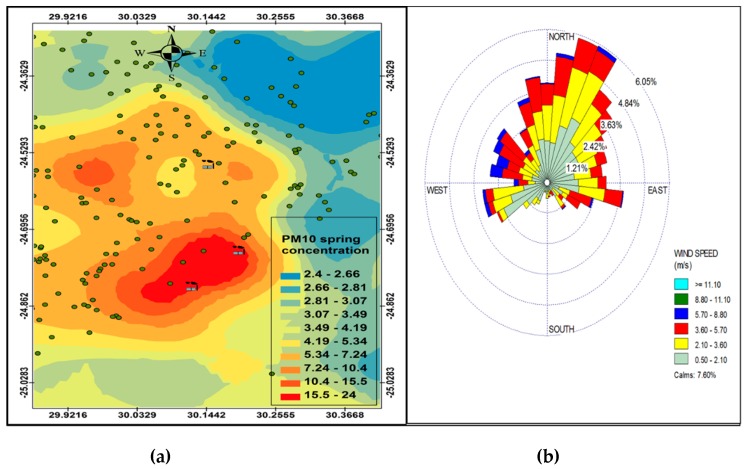
Spring variation in PM_10_ (µg.cm^−3^) ambient concentration (**a**) and wind roses (**b**).

**Figure 9 ijerph-16-03455-f009:**
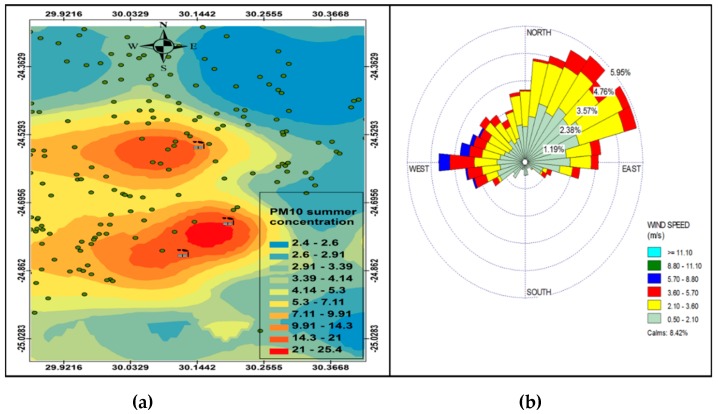
Summer variation in PM_10_ (µg.cm^−3^) ambient concentration (**a**) and wind roses (**b**).

**Figure 10 ijerph-16-03455-f010:**
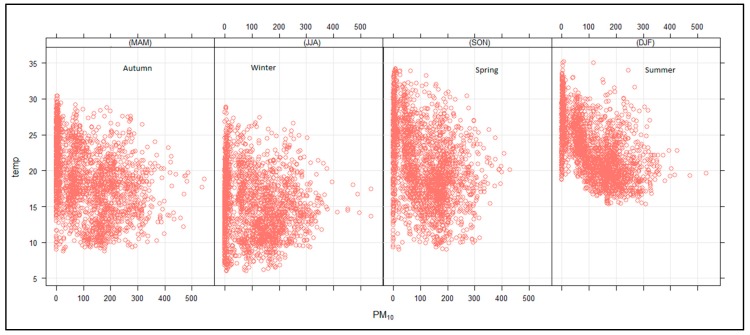
Relationship between PM_10_ (µg.cm^−3^) and ambient temperature (°C) for autumn, winter, spring and summer.

**Figure 11 ijerph-16-03455-f011:**
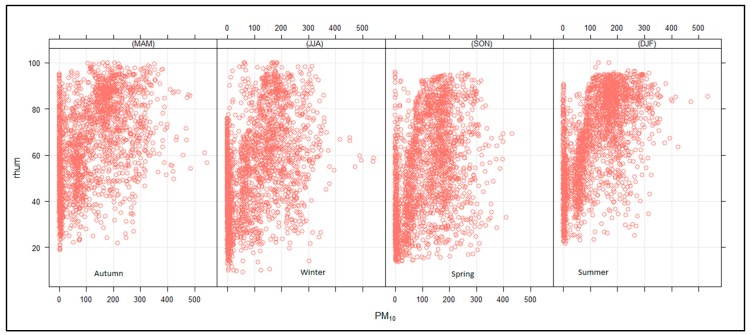
Relationship between PM_10_ (µg.cm^−3^) and relative humidity (%) for autumn, winter, spring and summer.

**Figure 12 ijerph-16-03455-f012:**
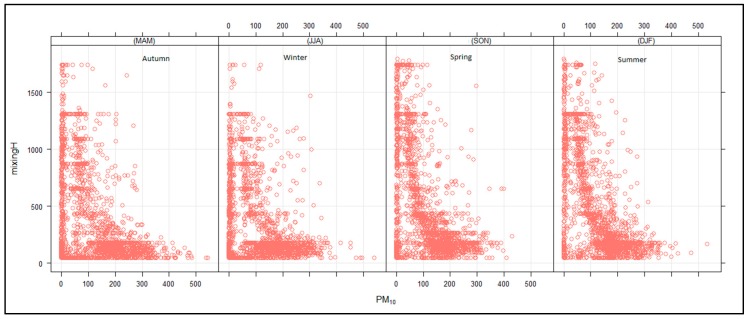
Relationship between PM_10_ (µg.cm^−3^) and mixing height (m) for autumn, winter, spring and summer.

**Figure 13 ijerph-16-03455-f013:**
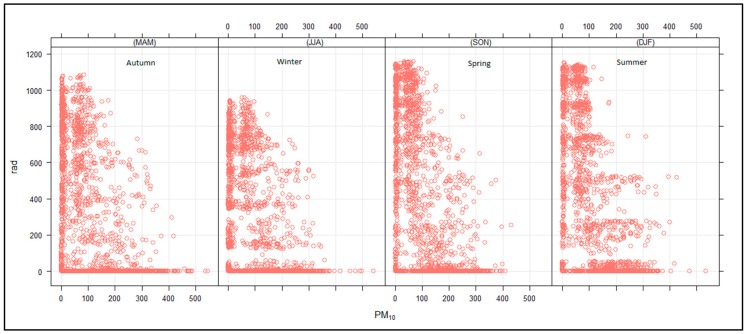
Relationship between PM_10_ (µg.cm^−3^) and solar radiation (KWh/m^2^) for autumn, winter, spring and summer.

**Table 1 ijerph-16-03455-t001:** Population statistics by age from the 2011 census for the GTM (statistics South Africa).

Age Group	Population *N* (%)
0–14	115,809 (34.5%)
15–64	202,748 (60%)
65+	17,119 (5.5%)
Total	335,676 (100%)

**Table 2 ijerph-16-03455-t002:** 2016 industrial emissions in the GTM.

Source	PM_10_ Emissions (kg/yr)	Stack Height (m)	Stack Diameter (m)	Exit Velocity (m/s)	Exit Temperature (°C)	Gas Flow Rate	Operating Hours
Smelter 3 (Stack 1)	1598	56.5	0.78	21.55	75	6.24	8760
Smelter 3 (Stack 2)	365	64.45	0.78	21.55	75	3.48	8760
Smelter 3 (Stack 3)	10,231.	64.45	0.78	21.55	75	3.96	8760
Smelter 2 (Stack 1)	11,416	32.0	1.6	24.13	137	5.2	8760
Smelter 2 (Stack 2)	2611	30.0	1.2	25.93	118	2.9	8760
Smelter 2 (Stack 3)	7308	30.0	1.7	25.18	110	3.3	8760
Smelter 1 (Stack 1)	13,699	50.3	1	24.13	160	5.72	8760
Smelter 1 (Stack 2)	3133	61.5	0.8	32.96	66	3.19	8760
Smelter 1 (Stack 3)	8769	60.6	1	30.59	50	3.63	8760

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
