# Peer review of "Spatial and Temporal Variation of PM10 from Industrial Point Sources in a Rural Area in Limpopo, South Africa"

_ijerph, 2019, doi:10.3390/ijerph16183455_

Round 1

Reviewer 1 Report

Please apply the following comments and recommendations:

- Page 3 Line 104, please, specify and justify the population classification. It is suitable consider two groups of population: child (<5 years old) and adults >65 years old. It would be better to include an additional class (children <5 years old).

Page 3 Line 111. Idicate the source of the database of parameters like soil properties, topography and deep soil

In Figure 3, in the map of spatial variation for PM10 concentrations, it would be better to put a degraded scale in colors (instead of the numbers in the concentrations isopleths) so that the concentrations are better visualized, since the numbers are not displayed well. Apply this comment to all the spatial distribution maps.

In page 5, Line 141 (Figure 3b), indicate the used software to construct the wind roses.

In 3.1 Section, it is necessary to include a health risk assessment (considering the population classification that you made at the begining of the paper) in order to know the non-cancer and cancer risk that could represent the found concentrations of PM10 in the study site.

In general, it is necessary to include other impacts:

The health risk assessment could cover health impacts since the exposure to PM section is very poor and does not provide additional information on the health impacts of exposure to particles (only consider the exceedances to the air quality standards).

If the authors, only are going to consider impacts on health, I suggest to change the title "Health impacts of point sources in an industrialized  rural area in Limpopo, South Africa" but only if the author include an improved section about health impacts (health risk assessment). On the contrary, I suggest to change the title "Spatial and temporal variation of PM10 in an industrialized rural area in Limpopo, South Africa" due to authors only considered the relation of PM10 concentration with meteorological parameters. 

Author Response

Reviewer 1

- Page 3 Line 104, please, specify and justify the population classification. It is suitable consider two groups of population: child (<5 years old) and adults >65 years old. It would be better to include an additional class (children <5 years old).

>>>The authors did not do the population distribution survey. The information was extracted from the Statistics South Africa website. Statistics South Africa collect information on age grouping to address the following;

·         To project the school age, working class, and elderly.

·         The provision of social services and life expectancy.

Therefore, there is a need for Environmental Department to engage with the Statistics Department to include the vulnerable groups (<5years old) in future surveys.

Page 3 Line 111. Indicate the source of the database of parameters like soil properties, topography and deep soil.

>>> We have added the source of the database to the manuscript in tracked changes.

In Figure 3, in the map of spatial variation for PM10 concentrations, it would be better to put a degraded scale in colours (instead of the numbers in the concentrations isopleths) so that the concentrations are better visualized, since the numbers are not displayed well. Apply this comment to all the spatial distribution maps.

>>>The maps were improved to include concentration scales. This has proved to be useful as the isopleths were hiding some information on the spatial variation

In page 5, Line 141 (Figure 3b), indicate the used software to construct the wind roses.

>>> We have added the software name to the manuscript in tracked changes.

In 3.1 Section, it is necessary to include a health risk assessment (considering the population classification that you made at the beginning of the paper) to know the non-cancer and cancer risk that could represent the found concentrations of PM10 in the study site.

>>> We changed the title of the manuscript and section 3.1

In general, it is necessary to include other impacts:

The health risk assessment could cover health impacts since the exposure to PM section is very poor and does not provide additional information on the health impacts of exposure to particles (only consider the exceedances to the air quality standards).

>>>We have improved the wording in the section.

If the authors, only are going to consider impacts on health, I suggest changing the title "Health impacts of point sources in an industrialized rural area in Limpopo, South Africa" but only if the author include an improved section about health impacts (health risk assessment). On the contrary, I suggest changing the title "Spatial and temporal variation of PM10 in an industrialized rural area in Limpopo, South Africa" due to authors only considered the relation of PM10 concentration with meteorological parameters.

>>>We thank the reviewer and we have amended the title to the second proposal.

Reviewer 2 Report

Major poiints:

Authors uses double standars of WHO and NAAQS. line 146, they say the highest annnual concentration was lower than NAAQS standard. But in line 281, they say that simulated value is smaller but comparable with the standard of WHO. This treatment is confusing.

This paper just show the simulated values. However, reliability of the calculation is not clear,andcomparison with observation data is required.

In my opinion, the simulation with not only one type of source but also with other sources sounds more.

Minor points:

All figures, too hard to see concentrations.

in Figure 1 a, smelters looks not in black, and  in 1 b, smelters are not shown in triangles.

In table 2, smelters are labeled wit A B and C. but in other, 1, 2, and 3 are la beled.

Author Response

Reviewer 2

Authors uses double standards of WHO and NAAQS. Line 146, they say the highest annual concentration was lower than NAAQS standard. But in line 281, they say that simulated value is smaller but comparable with the standard of WHO. This treatment is confusing.

>>> We have made appropriate changes to cover both the WHO guidelines and the South African NAAQS

This paper just show the simulated values. However, reliability of the calculation is not clear, and comparison with observation data is required.

>>> The paper is based on the simulation of PM10 concentrations due to non-availability of surface monitoring data in the area.

In my opinion, the simulation with not only one type of source but also with other sources sounds more.

>>> The study recommends that in future studies all major sources in the area should be included in the simulation.

Minor points:

All figures, too hard to see concentrations.

>>>We have amended all of the figures to ensure the concentrations can be more easily read.

In Figure 1 a, smelters looks not in black, and in 1 b, smelters are not shown in triangles.

>>> Figure 1a was generated using ArcScence in GIS 10.1. The ArcScene did not allow the display of Environmental icons, only general icons were showing on the map, hence the smelter could only by shown in triangles.

In table 2, smelters are labelled with A B and C. but in other, 1, 2, and 3 are labelled.

>>> We corrected the smelter naming with tracked changes in Table 2